# Dengue Virus Capsid Protein Facilitates Genome Compaction and Packaging

**DOI:** 10.3390/ijms24098158

**Published:** 2023-05-02

**Authors:** Priscilla L. S. Boon, Ana S. Martins, Xin Ni Lim, Francisco J. Enguita, Nuno C. Santos, Peter J. Bond, Yue Wan, Ivo C. Martins, Roland G. Huber

**Affiliations:** 1Bioinformatics Institute (BII), Agency for Science, Technology and Research (A*STAR), Singapore 138671, Singapore; 2Department of Biological Sciences (DBS), National University of Singapore (NUS), 16 Science Drive 4, Singapore 117558, Singapore; 3Instituto de Medicina Molecular, Faculdade de Medicina, Universidade de Lisboa, Av. Prof. Egas Moniz, 1649-028 Lisbon, Portugal; 4Genome Institute of Singapore (GIS), Agency for Science, Technology and Research (A*STAR), Singapore 138672, Singapore

**Keywords:** dengue, RNA structure, RNA–protein interactions, virus packaging

## Abstract

Dengue virus (DENV) is a single-stranded (+)-sense RNA virus that infects humans and mosquitoes, posing a significant health risk in tropical and subtropical regions. Mature virions are composed of an icosahedral shell of envelope (E) and membrane (M) proteins circumscribing a lipid bilayer, which in turn contains a complex of the approximately 11 kb genomic RNA with capsid (C) proteins. Whereas the structure of the envelope is clearly defined, the structure of the packaged genome in complex with C proteins remains elusive. Here, we investigated the interactions of C proteins with viral RNA, in solution and inside mature virions, via footprinting and cross-linking experiments. We demonstrated that C protein interaction with DENV genomes saturates at an RNA:C protein ratio below 1:250. Moreover, we also showed that the length of the RNA genome interaction sites varies, in a multimodal distribution, consistent with the C protein binding to each RNA site mostly in singlets or pairs (and, in some instances, higher numbers). We showed that interaction sites are preferentially sites with low base pairing, as previously measured by 2′-acetylation analyzed by primer extension (SHAPE) reactivity indicating structuredness. We found a clear association pattern emerged: RNA-C protein binding sites are strongly associated with long-range RNA–RNA interaction sites, particularly inside virions. This, in turn, explains the need for C protein in viral genome packaging: the protein has a chief role in coordinating these key interactions, promoting proper packaging of viral RNA. Such sites are, thus, highly consequential for viral assembly, and, as such, may be targeted in future drug development strategies against these and related viruses.

## 1. Introduction

Flaviviruses are responsible for a variety of severe diseases affecting populations in tropical and subtropical regions around the world [1]. The *Flavivirus* genus encompasses, among other, dengue virus (DENV), Zika virus (ZIKV), West Nile virus, Japanese encephalitis virus, and tick-borne encephalitis virus. Flaviviruses are vector-borne diseases with complex life cycles in arthropod and mammalian hosts [2,3,4]. The spectrum of symptoms from infection by flaviviruses encompasses hemorrhagic and neurological pathologies [5]. Interestingly, the molecular architectures of these viruses are very similar. All flaviviruses have an approximately 11 kb long (+)-sense single-stranded RNA genome with a single open reading frame that encodes a genome polyprotein which is post-translationally cleaved into the constituent viral proteins [6]. These proteins divide into structural proteins, which are present in viral particles, and non-structural proteins, which are expressed in infected cells [4,7]. The structural proteins comprise the envelope (E) and membrane (M) proteins, which together with a lipid bilayer make up the viral envelope [8,9,10]. These proteins are complemented by the highly cationic capsid (C) protein.

Flavivirus C proteins are approximately 100 amino acid residues in length and form conserved homodimeric quaternary structures [11,12,13]. Each monomer possess an intrinsically disordered N-terminal domain and four α-helices, α1 to α4 [11], with the dimer displaying an asymmetric charge distribution: one side contains a hydrophobic pocket (α2-α2′) alongside the disordered N-terminal domain, that is mostly involved in interactions with host lipid systems [14,15] and, possibly, the viral envelope, while the C-terminal side (α4-α4′) is highly positively charged and proposed to mediate C protein binding to viral RNA [16] (notwithstanding a similar role for other positively charged regions of the protein). Deletion of this critical C protein leads to the production of empty virus-like particles (VLPs), often in non-icosahedral geometries, comprising solely E and M proteins, without the viral genome [10,17]. Hence, it is clear that C proteins play a crucial role in the packaging of the viral genome and in the formation of infectious virions. As such, any fresh evidence on the molecular details of DENV and/or ZIKV C protein interaction with viral nucleic acids would be instrumental in gaining crucial new knowledge. However, whereas the structure of the envelope of other flaviviruses was studied in detail (using a variety of structural biology, biophysical, and computational techniques), this is not the case for DENV and ZIKV. The structure of the packaged genome and the role that C proteins play remained elusive. This is in part due to the C protein-RNA complex at the DENV and ZIKV virion center lacking a very clear regular geometry that facilitates experimental approaches and results interpretation [6,17,18,19,20]. With this in mind, we tried to contribute to generating a new understanding, by directly testing DENV C-RNA binding.

Briefly, in this study, we investigated the characteristics of C protein interactions with the genomic viral RNA. We determine the stoichiometry of the DENV C protein–RNA complexes by titrating the amount of proteins that could complex with the virus genome and localize the C protein binding regions in genomes that are in vitro refolded and in packaged genomes inside mature virions. Employing an approach based on a 2′-acetylation analyzed by primer extension (SHAPE) mapping [21] and taking advantage of our previous experience with ZIKV and DENV RNA SHAPE experiments [22], we found that C proteins preferentially interact with single-stranded regions of the viral RNA, as determined by comparing the binding patterns observed here with the data from those previous SHAPE experiments [22]. Briefly, SHAPE-MaP [23] enables the assessment of RNA secondary structure through chemical modification and profiling using high throughput sequencing. Double-stranded regions are protected from chemical modification, while single-stranded regions could be modified by a compound (NAI, 1H-Imidazol-1-yl(2-methyl-3-pyridinyl)methanone) to result in 2′-OH acylation. This acylation then causes errors in reverse transcription that can be quantified through sequencing [23]. We also establish that similar patterns are observed for DENV and for the closely related ZIKV C protein. Furthermore, we show that, particularly inside virions, DENV C protein interaction sites strongly correlate to sites that form long-range RNA–RNA interactions, as quantified by SPLASH [22,24,25]. Overall, these observations help to clarify and explain the role that C proteins play in the compaction and packaging of the viral genome. Such detailed understanding contributes to identify key binding spots, which may serve as potential targets for future drugs and/or therapies.

## 2. Results

### 2.1. The Number of C Proteins Bound to the RNA Genome Is Consistent with Early Estimates

To better understand how the C proteins bind along the viruses genomes, we first performed in vitro transcription of DENV2 genomic RNA (see Section 4: Materials and Methods). We then folded each genomic viral RNA separately and incubated them with their respective purified C protein, in solution, at RNA-to-protein monomer molar ratios of 1:20, 1:100, 1:250, and 1:500 (see Section 4: Materials and Methods). The C proteins form homodimers in solution and, hence, these ratios correspond to 10, 50, 125, and 250 C protein dimers per genomic RNA, respectively. For the sake of simplicity, as the dimer is the C protein functional unit in solution, from here onwards, when RNA:C protein ratios are discussed, they refer to RNA:C protein dimers (unless specifically stated otherwise). Recombinant C protein was produced as previously described [16,26] and was confirmed to have the expected size, secondary structure, and desired purity. Two replicates of incubates and controls containing no C protein were subjected to nuclease digestion and size filtered on agarose gel. Suitably sized fragments were ligated with sequencing adapters, amplified by PCR, and sequenced to reveal the regions protected from digestion by protein interactions (Figure 1A). Signal per million reads (SPMR) were determined for each position along the viral genomes. When normalizing for the distribution of reads obtained in the controls, binding locations may be inferred: RNA regions to which the C protein binds are protected from digestion and show increased abundance in C protein treated samples (compared to controls), which was indeed observed (Figure 1A). To determine the amount of C proteins needed to saturate binding to the virus genome, we calculated the distributions of reactivities as the protein amounts increase over the control in vitro (Figure 1B,C), performing a similar analysis for the in virion experiments (Figure 1D,E). As full charge complementarity between the viral genome and C protein would require approximately 125 C protein dimers per RNA molecule (depending on pH), we tested up to that value and the double of it (1:250 RNA:C protein dimer). It was apparent from analyzing the statistical significance between subsequent concentrations that saturation was already reached at the RNA:C protein dimer ratio of 1:125, and increasing the RNA:C protein ratio to 1:250 had only a negligible effect. Hence, it can be deduced that the native ratio of RNA to C protein dimer is somewhere between 1:50 and 1:125 (i.e., 100 to 250 C protein monomers), a level consistent and well within the interval of the number estimated for other structural proteins (roughly 1:90 RNA/protein dimer), as discussed ahead.

### 2.2. C Proteins Bind Specific RNA Genome Sections of Defined Nucleotide Lengths, as Single or Dimer Pairs

To map the C protein binding along the virus genome, we calculated the locations on the viral RNA molecule that showed an increased ratio of reads between control and C protein containing samples, both for the in vitro (Figure 1F) and the in virion (Figure 1G) experiments. From a basic observation of these data, it became immediately clear that DENV C protein, in both types of experiments (in vitro and in virion), binds some sections of the viral RNA very strongly but not as much to other regions. We then asked ourselves if there could be a motif for the binding or some particular GC content within the RNA sequence that could trigger the binding but, in fact, we did not observe a statistically significant difference in the distribution of nucleotides in the RNA binding regions. There was no clear pattern or difference when these were compared with the remaining regions of the genome: RNA-C binding is likely driven by electrostatic interactions with the RNA backbone both in vitro (Figure 2A) and in virion (Figure 2B). Then, we looked at the average length of the RNA:C interacting segments. This line of analysis proved to be very informing, as the RNA sections that bind the C protein showed a multimodal distribution consistent with clusters of fixed-length interactions (namely dimer singlets and dimer pairs), both in the in vitro (Figure 2C) and even more pronounced in virion (Figure 2D).

### 2.3. C Proteins Bind to the Viral Genome with a Preference for Structured RNA Regions

We analyzed the data further and, using a cutoff corresponding to the 75th percentile to localize binding regions, we found statistically significant evidence of a preference for single-stranded regions in the regions identified via the in vitro experiments (Figure 2E), but not in the in virion (Figure 2F) data. Notwithstanding, RNA-C protein in virion binding data SHAPE reactivity distribution patterns (Figure 2E). Next, taking advantage of previous work by us and others [22,24,25] on long-range RNA–RNA interactions, we evaluated the preference of the C protein for such regions. We found evidence of a preference for regions of the genome involved in long-range intramolecular RNA–RNA interactions: in the in vitro experiments, there was a very weak preference of the C protein for such sites (Figure 2G), which was clearly corroborated and very much evidenced, becoming statistically significant, when we analyzed the in virion experimental results (Figure 2H) in the same manner.

Considering only the data from the in virion experiments, it can be argued that as the protein and the RNA are obtained linked directly form the virion, and co-purified and digested together, the results are even more interesting. The preference for low-structure regions as measured by SHAPE was inverted and we observed a significant association with structured RNA (Figure 2F). Particularly striking was the shift regarding observed long-range intramolecular interactions, with sites that form strong interactions (as determined by SPLASH) considerably overrepresented in C protein binding sites (Figure 2H). Importantly, the multimodal pattern of interaction site lengths was preserved (Figure 2D) and appeared even more pronounced for the in virion data. Finally, to better compare in vitro and in virion data, a comparison of genomic locations of interaction sites was given (Figure 2I). It shows many conserved interactions, which indicates that the non-conserved interactions cause significant preference changes in binding behavior.

Overall, long-range RNA–RNA interaction spots were, thus, also clearly targeted by the capsid protein, a valuable insight into viral assembly mechanisms, as discussed ahead. To sum up the findings up to this point, first, it can be stated that there was clear selectivity of the C protein for specific viral RNA regions (Figure 1F). The mechanism by which this occurred is unclear but, possibly, at first, the RNA:C protein interaction is, at least in part, mostly driven by electrostatics forces (Figure 2A,B) and, following, as it matured and stabilized, the protein and the RNA self-organized into a more ordered multimodal binding pattern along the viral genome involving mostly singlets or paired dimers of the capsid protein (Figure 2C,D). There was a selectivity for some RNA sections (Figure 1F and Figure 2E,F) and, likely, the ability to initiate and/or model/stabilize long range interactions (Figure 2G,H).

These findings led us to investigate whether or not the C–RNA interaction in ZIKV displayed a similar pattern. To ascertain that, we performed titration of in vitro transcribed RNA for the ZIKV genome using ZIKV C protein. Similar patterns as for in vitro DENV C protein were observed and results are presented in the Appendix A. In fact, ZIKV C binding to its RNA likewise showed a strong preference for RNA sites of long-range intramolecular RNA–RNA interactions. Overall, this further supports the validity of the DENV C:RNA interaction findings.

## 3. Discussion

### 3.1. Specific C Protein Binding to RNA Is Observed and Within Expected Values

Our analysis of binding locations of C proteins on DENV RNA revealed distinct patterns for open and encapsulated states, with several commonalities and interesting differences. C protein plays a crucial functional role in the packaging process and our study provides some insight into the functioning of C protein in this role. First, the in vitro transcribed (IVT) RNA association patterns put the upper limit on the number of C proteins binding to a single genomic RNA at a molar ratio of about 1:250, which is consistent with the abundance of other structural proteins in a mature virion. A single virion includes 180 E and M proteins [10]. The mechanism of flavivirus translation does not allow for variations in protein ratios, as all viral proteins are translated as a single polyprotein and post-translationally cleaved. Hence, it is likely that the average ratio of RNA:C protein monomer is also close to 1:180. Such value would imply a 1:90 RNA:C protein dimer ratio, which is very consistent with our findings, supporting an interval of 1:50 to 1:125 RNA:C protein dimer. Moreover, such a ratio would also be consistent with the absence of a documented mechanism in Flaviviruses to regulate the expression of each viral protein independently. In contrast, such a mechanism does occur in other viruses, for example, via subgenomic coding segments are present in Chikungunya virus [27] or SARS-CoV-2 [28]. The observed ratio would also be consistent with full charge complementarity between vRNA and approximately 1:120 RNA:C protein dimers depending on local pH. Our study employed IVT vRNA, as a proxy for intracellular vRNA, alongside vRNA from virions (meaning that is also extracted from infected cells and already in complex with the C protein). Having two sources of vRNA (IVT vs. virion source) to study the association with the C protein enables to double-check the data obtained. For example, IVT could, in some instances, adopt other structure(s) besides the most common one, which could be described as the intracellular vRNA native structure. Recent work by Sanford et al. [29] shone light on this. The authors suggested that the circularization regions of ZIKV vRNA in the 5′ and 3′ UTRs adopt a linear conformation that is consistent with findings of a more linear structure within infected cells. Thus, the circular vs. linear conundrum may not apply, as both forms could be biologically relevant. Moreover, the SHAPE data presented in the study by Stanford et al. further corroborate this view, as it indicates a high correlation of IVT and native structures. Therefore, the vRNA can be in a linear form as opposed to a circularized one (although the later seems to be more abundant intracellularly, in some instances [22,24,25]). Thus, studying, as in our approach, the C protein association with IVT as well as with virion vRNA is highly informative of alternative vRNA conformations and, thus, modes of association. In any case, our data are consistent and support our general conclusions, and are in line with the overall expected findings, as detailed ahead.

### 3.2. C Protein Regulated Binding to Structure RNA Regions can Promote and/or Stabilize Viral Packaging

Initial association of C protein with RNA significantly favors single-stranded stretches in DENV (and ZIKV), while this pattern reverses in the packaged state, where a significant preference for double-stranded regions was observed. Interestingly, there were interaction sites shared between open and encapsulated states, thus indicating that this shift was occasioned by the non-conserved positions only observed in virion. This is consistent with a mechanism whereby the C protein facilitates, or at least does not impede, the formation of a compact RNA secondary structure in the mature virion. This finding is supported by the strong preference for regions of abundant long-range intramolecular RNA–RNA interactions in the virion, indicative of a prominent role for the C protein in the formation and/or stabilization of such interactions, particularly when contrasted with the weak preference for such regions in less spatially constrained environments.

### 3.3. C Protein Dimers Bind as Single Molecules or in Pairs to RNA and Can Promote Genome Compaction

Comparison of average genomic interaction lengths with the spatial dimensions of C protein indicate that stretches of approximately 15 nt would be sufficient to span the length of a single C protein homodimer. This was confirmed by the observed average interaction lengths both in the IVT and in virion experiments, which exhibited a peak at lengths of 15–20 nt. Interestingly, under spatially accommodating conditions, a multimodal distribution was observed, with a second peak at a length close to 40 nt, which indicates the clustering of two neighboring C protein homodimers. In virion, this multimodal structure was preserved and the data suggest the formation of higher order clusters with lengths round 60–75 nt (Figure 2D). This clustering of C proteins was also apparent in the comparison of interaction sites and lengths of IVT and in virion experiments (Figure 2I), showing less but longer interactions. We postulate that the formation of C protein clusters along the genome, combined with the formation of long-range intramolecular RNA–RNA interactions at these sites, may be a crucial driver of the necessary compaction of the genomic RNA that is a precondition for packaging. It would appear, however, that the packaging process in flaviviruses is less regular than the ‘beads-on-a-string’ packaging observed, for instance, in influenza viruses [30]. Such a role would be consistent with previous findings of chaperone activity for the N-terminal disordered region of C proteins [31]. These key genomic regions involved in interactions with the C protein may, thus, be of pharmacological interest for future drug design strategies.

## 4. Materials and Methods

### 4.1. DENV and ZIKV C Proteins Expression and Purification

Following previously described procedures [16], DENV and ZIKV C proteins were expressed in *Escherichia coli* cells C43 (DE3) and C41 (DE3), transformed with the DENV 2 C protein gene (GenBank database: AAC59275.1) and with the ZIKV Brazil C protein gene (GenBank database: KU497555.1), respectively, cloned into a pET21a plasmid (ampicillin resistance). An amount of 6 mL lysogeny broth (LB) medium with 100 μg/mL ampicillin were inoculated with a single *E. coli* colony, containing the pET21a, from LB-agar plates with 100 μg/mL ampicillin, which was then incubated overnight at 220 rpm, 37 °C. The overnight culture was used to inoculate LB with 100 μg/mL ampicillin, to a final dilution of 1:50. The main culture was incubated at 220 rpm and 37 °C, until the optical density at 600 nm (OD600) reached 0.8. Protein expression was induced with 0.5 mM isopropyl β-D-1-thiogalactopyranoside (IPTG), at 25 °C overnight and with 220 rpm agitation. The overnight culture was centrifuged at 7000× *g* for 40 min at 4 °C. Supernatants were discarded and the cell pellets were resuspended in 1/10 of the volume of the initial culture, in a buffer with 25 mM HEPES, pH 7.4, 0.2 M NaCl, 1 mM EDTA, 5% (*v*/*v*) glycerol (Buffer 0.2 M NaCl) and 10 μM protease inhibitor cocktail (Sigma-Aldrich, St. Louis, MO, USA). Cells were lysed by sonication, keeping the suspension on ice, with 5 to 7 pulses (1.5 min) and pauses of 2 min. NaCl was added to the cell lysate to achieve a final concentration of 2 M, and incubated for at least 1 h, with agitation, at 4 °C. The suspension was centrifuged at 16,100× *g* for 30 min at 4 °C, and supernatants were collected. Before the first chromatography step, these supernatants were diluted fourfold with Buffer 0.2 M NaCl. The diluted supernatants were applied to a 5 mL HiTrap heparin column (HiTrapTM Heparin HP, GE Healthcare, Little Chalfont, UK) coupled to an ÄKTA purifier system (ÄKTA explorer, GE Healthcare, Little Chalfont, UK). The C protein was then eluted, performing a NaCl gradient from 0.2 to 2 M. The fractions corresponding to the C protein were collected and injected in a size exclusion column (S200). C proteins were purified in a buffer with 55 mM KH_2_PO_4_ and 550 mM KCl, pH 6.0. DENV C protein purified fractions were concentrated with a Centriprep Centrifugal Filter (Millipore; 3 kDa cutoff) and stored at −80 °C. ZIKV C protein purified fractions were centrifuged at 13,400× *g* to remove protein aggregates and the supernatants were stored at −80 °C. The purity of the purified recombinant C protein samples was assessed by 15% sodium dodecyl sulfate–polyacrylamide gel electrophoresis (SDS-PAGE). Protein concentrations were determined from the sample absorbance at 280 nm, with extinction coefficients calculated from their primary structures. Matrix-assisted laser desorption/ionization, time-of-flight mass spectrometry (MALDI-TOF MS) analysis showed one peak, with the expected mass of the protein monomer and much lower peaks corresponding to a very low degradation. Unless otherwise stated, all chemicals were obtained from Sigma (Sigma-Aldrich, St. Louis, MO, USA).

### 4.2. Circular Dichroism

The secondary structure of the C proteins was confirmed by circular dichroism (CD) spectroscopy. CD measurements were carried out in a JASCO J815 (Tokyo, Japan), using 0.1 cm path length quartz cuvettes with 220 μL total volume, data pitch of 0.5 nm, velocity of 200 nm/min, data integration time of 1 s and performing 3 accumulations. Spectra were acquired in the far UV region, between 200 and 260 nm, with 1 nm bandwidth. The temperature was controlled by a JASCO PTC-423S/15 Peltier equipment, at 25 °C. C protein concentration was 20 μM (monomer) in 50 mM KH_2_PO_4_, 200 mM KCl, pH 7.5. Spectra were smoothed through the means-movement method (using 5 points) and normalized to mean residue molar ellipticity, [θ] (in deg cm^2^ dmol^−1^ Res^−1^). ZIKV and DENV C CD spectra corresponded to high α-helical structure contents, as expected and previously reported [11,12,14,16,26].

### 4.3. vRNA Structure Probing and C Protein-vRNA Digestion

Eight reactions, four for DENV and four for ZIKV, with 1 µg of vRNA in nuclease-free water were prepared in 0.2-mL PCR tubes. To renature the viral RNA, tubes were heated at 90 °C, for 2 min, in a thermal cycler with heated lid on and then transferred to ice for 2 min. 5 μL of 10 × RNA structure buffer were added to the vRNA and gently homogenized by pipetting up and down. Then, tubes were placed into the thermal cycler and temperature increased from 4 °C to 37 °C over 35 min. DENV or ZIKV C proteins were added to the four reactions in increasing concentrations, to achieve different final ratios of C protein monomers to one molecule of vRNA (1:20, 1:100, 1:250 and 1:500). Samples were incubated at 37 °C for 30 min, to allow the interaction of the C protein with vRNA. An amount of 5 μL of RNase A/T1 Mix (Rnase A 500 U/mL and Rnase T1 20,000 U/mL; Ambion AM2286, Thermo Fisher Scientific, Waltham, MA, USA) diluted 100 times were added to each tube, and were incubated at 37 °C for 30 min. The dilution of Rnase A/T1 Mix was previously determined with four serial 10 fold dilutions of Rnase A/T1 Mix from the stock solution. A total of 45 µL of nuclease-free water was added to each sample to adjust the final volume of the reaction mixtures to 100 µL, and then transferred to 1.5 mL microfuge tubes with the same volume of phenol–chloroform–isoamyl alcohol. After mixing, tubes were centrifuged at 13,200× *g* for 10 min at 4 °C. The aqueous phase of each sample was transferred to a new tube containing 10 µL sodium acetate 3 M and 1 µL glycogen 20 mg/mL. Then, 300 µL of ethanol 100% was mixed with each aqueous solution and, to precipitate the vRNA, samples were stored at −20 °C overnight. To collect the vRNA, samples were spin at 13,200× *g* for 30 min at 4 °C. Supernatants were discarded and 800 µL of 70% ethanol were added to the pellet. Samples were spun at 13,200× *g* for 15 min at 4 °C and the supernatants removed. A dry spin at 13,200× *g* for 1 min at 4 °C was made to completely remove ethanol. vRNA pellets were resuspend in 15 µL of nuclease-free water and stored at −80 °C.

### 4.4. Library Construction

Unless otherwise stated, all chemicals were obtained from Sigma (Sigma-Aldrich, St. Louis, MO, USA). An amount of 2 µL of 10× T4 polynucleotide kinase (PNK) buffer and 2 µL of T4 PNK were added to 14 µL of each vRNA sample. Reaction mixtures were incubated for 4 h at 37 °C. Then, an additional 0.5 µL of T4 PNK and 2 µL of 10 mM ATP were added, and the mixture reactions were incubated for 1 h at 37 °C. The final volume of the mixtures was adjusted to 50 µL by adding nuclease-free water. Then, high-quality vRNA samples were prepared with an RNA Clean & Concentrator kit (RNA Clean & Concentrator-5, Zymo Research, Irvine, CA, USA). Briefly, 100 μL of RNA binding buffer was added to each 50 μL of vRNA sample. Then, an equal volume of 100% ethanol was added and mixed. Samples were transferred to a column in a collection tube and centrifuged for 30 s at 12,000× *g*. The flow-through was discarded and 400 μL of the second RNA buffer were added. Viral RNA bound to the columns were washed twice. A dry spin was made to completely remove the washing buffer. Columns were transferred into RNA-free tubes and the vRNA was eluted by adding 8 μL of RNase-free water to the column matrix. After 1 min incubation, columns were centrifuged for 30 s at 12,000× *g*. The recovered vRNA was double eluted and samples were stored at −80 °C. Library preparation was made with a NEBNext Multiplex Small RNA Library Prep Set for Illumina kit (New England Biolabs, Ipswich, MA, USA), with small modifications to the protocol. 

An amount of 1 μL of 3′ SR Adaptor was added to 6 μL of the high-quality vRNA samples prepared, in nuclease-free PCR tubes. Reaction mixtures were incubated in a preheated thermal cycler for 2 min at 70 °C, and then transferred to ice. Then, 10 μL of 3′ Ligation Reaction Buffer and 3 μL of 3′ Ligation Enzyme Mix were added to each reaction mixture. Samples were incubated in a thermal cycler for 1 h at 25 °C. To prevent the formation of dimers of the remaining 3′ SR Adaptor, 4.5 μL of nuclease-free water and 1 μL of SR reverse transcription (RT) Primer for Illumina were added, and samples were incubated for 5 min at 75 °C, 15 min at 37 °C and 15 min at 25 °C.

For the ligation of the 5′ SR Adaptor, an adaptor mix was prepared. For that, 5′ SR Adaptor was incubated in a thermal cycler at 70 °C for 2 min, and then, immediately transferred to ice. Then, 1 μL of the denatured 5′ SR Adaptor was mixed with 1 μL of 5′ Ligation Reaction Buffer and 2.5 μL of 5′ Ligation Enzyme Mix. The adaptor mix was added to the samples with 3′ SR Adaptor and incubated in a thermal cycler for 1 h at 25 °C.

To perform the RT of the adaptor ligated vRNA samples, 30 μL of these samples were mixed in a nuclease-free tube with 8 μL of First Strand Synthesis Reaction Buffer, 1 μL of Murine RNase Inhibitor and 1 μL of ProtoScript II Reverse Transcriptase. Reaction mixtures were incubated at 50 °C for 60 min. Then, RT reactions were inactivated at 70 °C for 15 min and stored at −20 °C.

A large-scale PCR amplification was performed using 12 PCR cycles, the number of cycles identified as enough to construct a library. For that, 10 μL of the cDNA were mixed, on ice, with 2.5 μL of F-primer (SR primer), 2.5 μL of R-primer (with different barcodes for each sample), 5 μL of nuclease-free water and 20 μL of 2X Q5 Master Mix. The barcoded cDNA samples were normalized to 1000 ng/mL and sent to next generation sequencing on the Illumina platform.

### 4.5. Propagation of DENV

DENV serotype 2 (EDEN virus 3295, Genbank ID: EU081177) was amplified in C6/36 cells and harvested from the cellular media 4–5 days post-infection. Freshly collected virus was centrifuged at 14,000× *g* for 1 h at 4 °C to remove cellular debris. The virus supernatant was precipitated with PEG 8000 overnight, at 4 °C, to concentrate the viral particles. The precipitated virus was centrifuged at 14,334× *g* for 1 h at 4 °C and the pellet was further purified through 24% sucrose cushion. After sucrose cushion, the pellet was left to soften overnight at 4 °C in phosphate-buffered saline (PBS) at pH 7.4. The purified virus pellet was re-suspended and centrifuged at 13,000× *g* for 5 min to remove cellular debris. The purified virus supernatant was used immediately for the downstream experiments.

### 4.6. DENV RNA-Capsid Immunoprecipitation and Library Preparation

Unless otherwise stated, all chemicals and materials were obtained from Sigma (Sigma-Aldrich, St. Louis, MO, USA). Half of the purified DENV was kept as a control without crosslinking (Figure 1D, Control), while the remaining half of the virus sample was cross-linked at 254 nm, 400 mJ/cm^2^, on ice (Figure 1D, Crosslinked). Both the cross-linked and the non-cross-linked viruses were then lysed in radio-Immunoprecipitation assay (RIPA) buffer (50 mM Tris-HCI pH 7.6, 150 mM NaCl, 1% NP-40, 0.5% sodium deoxycholate, 0.1% SDS), supplemented with protease inhibitor cocktail (Merck, Darmstadt, Germany) and SUPERase-in (Thermo Fisher Scientific, Waltham, MA, USA), at 37 °C for 30 min, with constant shaking. The non-cross-linked samples were separated into two reactions where: (1) we directly extracted the RNA using TRIzol LS reagent (Thermo Fisher Scientific, Waltham, MA, USA), following the manufacturer’s instructions; (2) we proceeded with immunoprecipitation with the cross-linked samples. The non-cross linked control TRIzol RNA sample was fragmented with RNaseA/T1 cocktail mix (Thermo Fisher Scientific, Waltham, MA, USA), at 37 °C for 30 min. The reaction was inactivated by phenol/chloroform/isoamyl alcohol 25:24:1 (Sigma-Aldrich, St. Louis, MO, USA) and purified using the Zymo RNA Clean & Concentrator-5, according to the manufacturer’s instructions. The purified RNA was then used as the input for the cDNA library generation using the NEBNext Multiplex Small RNA library Prep Kit (New England Biolabs, Ipswich, MA, USA), according to the manufacturer’s instructions.

The cross-linked and non-cross-linked lysates were incubated with the dyna G beads-DENV2 anti-capsid antibody complex at 4 °C for 2 h, with constant shaking. After antibody binding, RNA was fragmented with RNaseA/T1 cocktail mix at 37 °C for 30 min. The fragmented complexes were washed four times with cold IP wash buffer (500 mM NaCl, 25 mM Tris pH 7.2, 0.05% Tween 20) on a magnetic rack. Samples were treated with proteinase K (Thermo Fisher Scientific, Waltham, MA, USA) at 37 °C for 1 h, to release the RNA fragments from the immuno-precipitated complexes, and RNA was extracted using TRIzol LS reagent. The extracted RNA samples were treated with T4 PNK (New England Biolabs, Ipswich, MA, USA) at 37 °C for 5 h and purified using the Zymo RNA Clean & Concentrator-5. The purified RNA was then used as input for the cDNA library generation using the NEBNext Multiplex Small RNA library Prep Kit.

### 4.7. Computational Analysis

Sequencing results from all samples were processed with ‘cutadapt’, to remove Illumina sequencing adapters, and subsequently aligned against the respective reference genomes, using bowtie2 with the ‘very-sensitive-local’ settings. The number of aligned reads for each sample is presented in Appendix A. Unaligned reads were discarded and aligned reads normalized to obtain signal per million reads (SPMRs) at each position of the viral genomes. Correlations of SPMR between replicates were determined and are shown in Appendix A. Replicates showed Pearson’s r correlations greater than 0.8. Consequently, samples were pooled for subsequent analysis. In order to assess increasing relative abundance, SPMR at each position was divided by SPMR for the control (IVT) or background (crosslinking) and the Log10 SPMR ratio for each position was calculated.

Significance of changes in SPMR and Log10 SPMR ratios were determined using a Mann–Whitney U test and are shown for all pairwise comparisons. Binding regions were defined as the 75th percentile of Log10 SPMR in the 1:500 samples, resulting in cutoffs of 2.2 and 0.5 for DENV and DENV crosslinking experiments, respectively. Subsequently, binding regions were characterized with respect to their nucleotide composition, and the divergence from non-binding regions was assessed with a χ^2^ test.

Similarly, binding and non-binding regions were compared with respect to their SHAPE reactivity and propensity for intramolecular long-range interactions, as determined previously by SPLASH [22]. The average length of interacting regions was determined and the distribution of lengths is given as histograms.

## Figures and Tables

**Figure 1 ijms-24-08158-f001:**
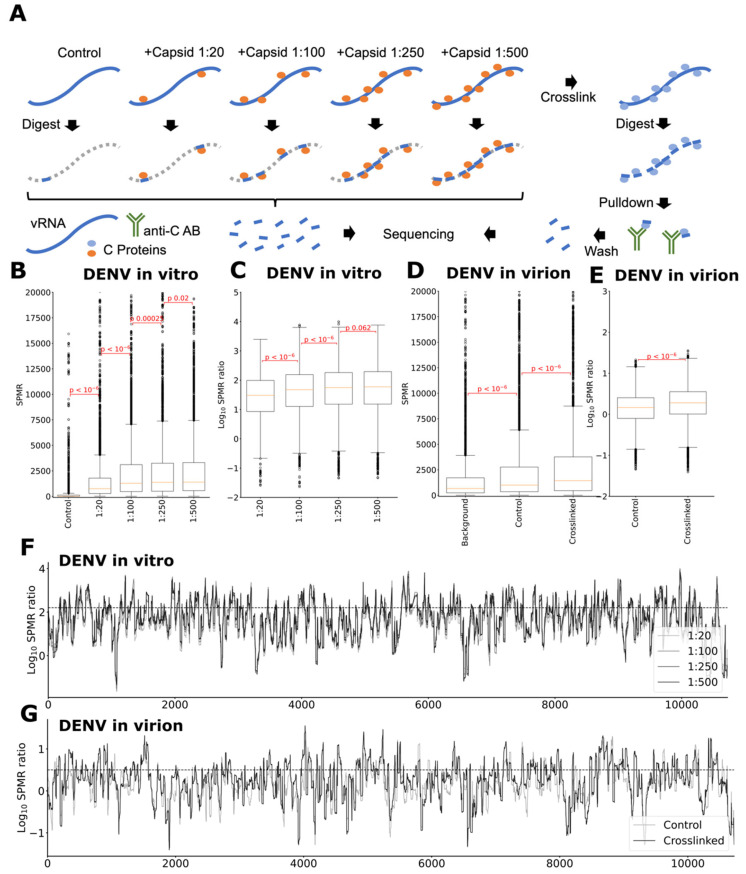
Footprinting of C protein binding on DENV genomic RNA. (**A**) Two distinct experiments to determine C protein interaction locations with genomic RNA were conducted on in vitro transcribed RNA and inside mature virions. (**B**) Abundance of reads after SPMR normalization in DENV and (**C**) ratio of reads abundance over control show that at a molar ratio of 1:250 the genomes are saturated with C protein, indicating that the saturation point lies at or below this level. (**D**) highlights absolute signal observed for crosslinking in virion and (**E**) indicates ratios for control and crosslinked experiments over background. Localization of the binding signal along the DENV genome is shown in (**F**) for the in vitro transcribed genomic RNA and (**G**) for the in virion crosslinked experiment.

**Figure 2 ijms-24-08158-f002:**
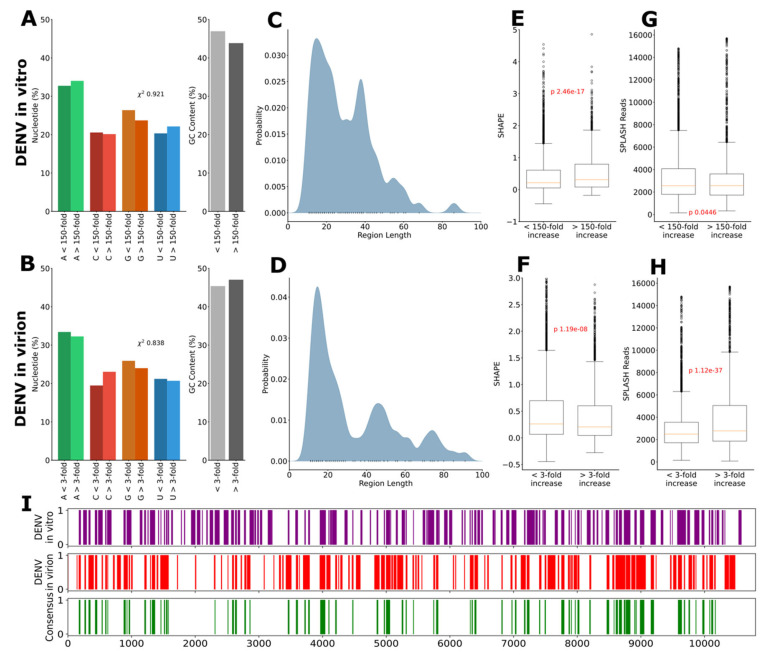
Characteristics of DENV C protein binding locations. The nucleotide composition and GC content of C protein binding locations are not significantly different from non-binding locations, supporting the notion that binding is largely driven by electrostatic interactions with the RNA backbone, both in vitro (**A**) and in virion (**B**). The average length of interacting segments shows a multimodal distribution consistent with clusters of fixed-length interactions in vitro (**C**) and even more pronounced in virion (**D**). Whereas binding in vitro preferentially occurs in regions of higher SHAPE reactivity, indicating a preference for single-stranded, open regions of the viral genome (**E**), this pattern is reversed in virion (**F**). Likewise, in vitro preference for regions of long-range intramolecular RNA–RNA interactions, as measured by SPLASH, is weak in vitro (**G**) but highly significant in virion (**H**). Interestingly, despite significant changes in the structure preference, several sites are shared between in vitro and in virion C protein binding, indicating that the shift is occasioned specifically by regions exclusive to in virion binding (**I**).

## Data Availability

Not applicable.

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
