# Peer review of "Dengue Virus Capsid Protein Facilitates Genome Compaction and Packaging"

_ijms, 2023, doi:10.3390/ijms24098158_

Round 1
Reviewer 1 Report
Flaviviruses, including Dengue and Zika virus, are responsible for a significant amount of disease in the human population. Packaging of the viral RNA is dependent upon interaction with the Capsid protein (C) although the molecular mechanisms underlying this dependency are not known. The study by Boon et al examines the binding characteristics of C protein with in vitro transcribed and virion RNA using footprinting and deep sequencing. Initial experiments determine reveals the binding stoichiometry of C protein on in vitro transcribed RNA. Analysis of protected regions did not reveal a string consensus suggesting it is driven primarily by charge:charge interactions. Interestingly, binding shows a preference for a distinct length of RNA. Comparison with SHAPE date generated from work previously published by this group reveals a distinct binding preference for C protein to single stranded regions of in vitro transcribed vRNA while virions show preference for structured regions. A similar analysis using SPLASH data from prior work, shows that C protein has a strong affinity for regions involved in long-range interactions in virions that isn’t reproduced with in vitro transcribed RNA. Comparison of in vitro and in virion binding sites reveals conserved and non-conserved regions.
Supplemental data showing analysis of in vitro transcribed ZIKV RNA shows similar stoichiometry of C protein binding and a preference for single stranded regions with certain size constraints and a preference for long-range interaction sites.
General Comments:
Characterizing the molecular details of how the C protein interacts with vRNA and its role in packaging are clearly important for better understanding these important human pathogens. This study presents convincing data regarding the stoichiometry of C protein binding to vRNA. While the authors discuss how this is consistent with the known amounts of E and M proteins found in virions this finding could be supported by experimental efforts to quantify C protein in virions. Unfortunately, it is less clear what to make of the conflicting findings regarding binding of C protein to in vitro transcribed vs virion RNA. The author’s present the interesting idea that in vitro binding is analogous to binding to vRNA prior to packaging, however no experiments or references are presented to support this hypothesis. Perhaps isolating vRNA from infected cells and incubating with C protein might help to answer this question. Another outstanding question is how to explain the lack of consistency between the protected fragment sizes found between in vitro transcribed and virion RNA. While it’s possible this is due to differences in the folded structures of vRNA vs that made in vitro, it might also suggest some inherent difference in the characteristics of bacterially produced C protein and that made in infected cells. The inclusion of the Zika virus data without fully integrating it into the Dengue analysis also muddies the waters a bit as it shows a strong association with long range interactions. Including an analysis of Zika virion RNA might confirm this and add support to the Dengue virion RNA analysis.
Specific Comments:
Line 104: How was secondary structure of purified C protein determined?
Figure 1D and E: Define background and control for the in-virion assays. How was this similar or not to experiments shown in 1B and C?
Reviewer 2 Report
Given the importance of the dengue virus in human health, authors analyzed the interactions of the C proteins with the viral genomic RNA, in solution and inside mature virions, via footprinting and cross-linking experiments. Authors indicate that C protein interacts with viral genomes at an RNA:C ratio below 1:250. Additionally, they found a clear association pattern for the RNA-C protein binding sites, which are strongly correlated with long-range RNA-RNA interaction sites, particularly inside virions.
The results are interesting; however, there are some aspects that should be included in the manuscript to support authors’ conclusions:
1. Is there any reason to use the C protein from DENV serotype 2 instead of other serotypes? Is there any difference with other DENV serotypes?
2. Is there any difference is the interaction of the C protein from one DENV serotype with the viral RNA from another DENV serotype?
3. It has been described that C protein from some flavivirus is present in the nucleus of infected cells as well as in lipid droplets. How these locations can alter or are required for viral packing?
Round 2
Reviewer 1 Report
The authors have responded to most of my criticisms and suggestions. They have indicated that technical issues prevented them from including analysis of ZIKV C protein:vRNA interaction. They have also included circular dichroism spectroscopy analysis of their C protein showing that it behaves as expected structurally although they have not included this data in the manuscript. Here I would like to suggest that since interpretation of the data presented requires that readers have confidence in the structural integrity of the C protein, this should be included in this manuscript. They have also added text explaining the background and control parameters for the in vitro assay.
Unfortunately, they seem to have misinterpreted my other comments. I was not trying to argue that in vitro transcribed RNA shouldn’t be used as a proxy for what’s going on inside the cell (or virion), I agree that is well established with appropriate caveats of course.
The response fails to address my comment that ‘While the authors discuss how this is consistent with the known amounts of E and M proteins found in virions this finding could be supported by experimental efforts to quantify C protein in virions’. Is there a reason these experiments are not technically possible or beyond the scope of the present study?
In the revised manuscript Lines 215-6 seem to suggest that in vitro transcribed RNA was isolated from infected cells.
Author Response
The authors have responded to most of my criticisms and suggestions. They have indicated that technical issues prevented them from including analysis of ZIKV C protein:vRNA interaction. They have also included circular dichroism spectroscopy analysis of their C protein showing that it behaves as expected structurally although they have not included this data in the manuscript.
Here I would like to suggest that since interpretation of the data presented requires that readers have confidence in the structural integrity of the C protein, this should be included in this manuscript. They have also added text explaining the background and control parameters for the in vitro assay.
We appreciate the Reviewer time and effort and the acknowledgement that the explanations concerning the technical issues previously raised, background and control experiments, are adequate and that those questions have been fully addressed. At the Reviewer suggestion, the circular dichroism data was included as a supporting figure in the revised Supplementary Information file (Figure S2), showing the protein structural integrity.
Unfortunately, they seem to have misinterpreted my other comments. I was not trying to argue that in vitro transcribed RNA shouldn’t be used as a proxy for what’s going on inside the cell (or virion), I agree that is well established with appropriate caveats of course.
We agree with the reviewer here.
The response fails to address my comment that ‘While the authors discuss how this is consistent with the known amounts of E and M proteins found in virions this finding could be supported by experimental efforts to quantify C protein in virions’. Is there a reason these experiments are not technically possible or beyond the scope of the present study?
We propose that our titration effort allows us to quantify the stoichiometry of the C protein binding to vRNA to a native ratio of RNA to C protein dimer somewhere between 1:50 to 1:125 (i.e., 100 to 250 C protein monomers), a level consistent and well within the interval of the number estimated for other structural proteins (roughly 1:90 RNA:protein dimer).
We agree that this result does not yield an exact ratio and that our observation that this region would be consistent with either identical C protein numbers to E and M proteins in virions (180 monomers, determined by cryo-EM) up to a maximal quantity of full charge complementarity (~240 monomers, depending on local pH as the charge of C protein might vary slightly under different pH conditions). We added a sentence raising this possibility to the discussion section. The given result is accurate to the limit of what we could accomplish within our experiments.
An accurate determination of the exact ratios would require a more extensive study of isolated and purified virions with subsequent fractionation of the different protein components, which would constitute a considerable additional effort beyond the scope of our study. We consider the structural patterns of C protein binding and the association with vRNA structural features to be the core result of our work.
In the revised manuscript Lines 215-6 seem to suggest that in vitro transcribed RNA was isolated from infected cells.
We removed this section in parenthesis as it was formulated in a misleading way. The vRNA was indeed in-vitro transcribed (and is thus isolated from C protein which would be present if intracellular vRNA would have been extracted). The word ‘extracted’ was a language mistake on our part, we meant to state that it is transcribed in the absence of C protein.
Reviewer 2 Report
The manuscript is now suitable for publication
Author Response
We thank the reviewer for their time and effort.